# Comprehensive Analysis of Imipenemase (IMP)-Type Metallo-β-Lactamase: A Global Distribution Threatening Asia

**DOI:** 10.3390/antibiotics11020236

**Published:** 2022-02-11

**Authors:** Pisut Pongchaikul, Paninee Mongkolsuk

**Affiliations:** 1Faculty of Medicine Ramathibodi Hospital, Chakri Naruebodindra Medical Institute, Mahidol University, Samut Prakan 10540, Thailand; phaninae@gmail.com; 2Integrative Computational BioScience Center, Mahidol University, Nakhon Pathom 73170, Thailand; 3Institute of Infection, Veterinary and Ecological Sciences, University of Liverpool, Liverpool L69 3BX, UK

**Keywords:** β-lactamase, carbapenemase, antimicrobial resistance

## Abstract

Antibiotic resistance, particularly beta-lactam resistance, is a major problem worldwide. Imipenemase or IMP-type metallo-β-lactamase (MBL) has become a more prominent enzyme, especially in Asia, since it was discovered in the 1990s in Japan. There are currently 88 variants of IMP-type enzymes. The most commonly identified variant of IMP-type enzymes is IMP−1 variant. IMP-type MBLs have been detected in more than ten species in Enterobacterales. Pseudomonas aeruginosa is the most frequent carrier of IMP-type enzymes worldwide. In Asia, IMP-type MBLs have been distributed in many countries. This work investigated a variety of currently available IMP-type MBLs at both a global level and a regional level. Out of 88 variants of IMP-type MBLs reported worldwide, only 32 variants were found to have susceptibility profiles. Most of the bacterial isolates carrying IMP-type MBLs were resistant to Carbapenems, especially Imipenem and Meropenem, followed by the 3rd-generation cephalosporins, and interestingly, monobactams. Our results comprehensively indicated the distribution of IMP-type MBLs in Asia and raised the awareness of the situation of antimicrobial resistance in the region.

## 1. Introduction

Multidrug resistance organisms, especially β-lactamase-harbouring pathogens, are a major global public health problem worldwide resulting in high mortality, high morbidity and rising economic costs [1]. The β-lactamase enzyme, which can be produced by both gram-positive bacteria and gram-negative bacteria, inactivates β-lactam antibiotics (i.e., penicillin, cephalosporin, carbapenem and monobactam) by hydrolysing the amide bond of β-lactam ring [2]. Currently, there are more than 7270 enzymes available in the β-lactamase database (Beta-Lactamase database. Available online: www.bldb.eu, accessed on 30 November 2021). β-lactamase can be classified into four classes based on Ambler classification. Class A, C, D include serine protease-derived β-lactamases while class B includes the metallo-or zinc dependent β-lactamase (MBL) [3].

Imipenemase (IMP) is encoded by *bla_IMP_* genes. Along with other enzymes in this group: Verona Integron-encoded Metallo-β-lactamase (VIM), São Paulo metallo-β-lactamase (SPM) and German imipenemase (GIM). IMP belongs to class B β-lactamase and has carbapenemase activity [4]. Similar to other MBLs, IMP MBL breaks β-lactam ring with zinc as a catalyst and the enzyme can be inhibited by EDTA. IMP is commonly transferred between organisms, especially Gram-negative bacteria, via class 1 or class 3 of integron [5]. The discovery of *bla**_IMP_**_−_**_1_* was first reported in Japan in 1988 from *P. aeruginosa* strain GN17203 [6]. There are currently 88 variants of IMP reported worldwide.

Even though IMP-type MBLs are important and widely distributed around the world, a comprehensive review of this enzyme has not been conducted. Moreover, a previous phylogenetic construction was restricted due to the limited number of available sequences. To understand the comprehensive picture of the *bla_IMP_* gene, a review of relevant literature and a phylogenetic tree reconstruction was performed to investigate the distribution of IMP-type MBLs, phylogenetic relationship of the genes, and the association between phylogenetic cluster and antibiotic susceptibility.

## 2. Materials and Methods

### 2.1. Review of Literature

A comprehensive literature search was performed by PM and PP on Pubmed/Medline and EMBASE until 30 November 2021 to obtain relevant articles. The search terms used were “IMP and β-Lactamases”. A list of references was stored and the duplicates were removed using Endnote. PM and PP separately screened and selected the titles and the abstracts mentioning IMP metallo-β-lactamase. Articles were included when the prevalence of *bla_IMP_* gene was reported. Articles were excluded when the English version was not available.

### 2.2. bla_IMP_ Gene Sequence Retrieval and Analysis

A total number of 88 sequences of IMP-type metallo-β-lactamase genes (*bla_IMP_*) were found and downloaded from both β-lactamase databases [7] (last accessed, November 2021) and GenBank database in November 2021. IMP−36, IMP−50 and IMP−57 could not be found and retrieved from both databases. Multiple sequence alignment of both nucleotide sequences and amino acid sequences was processed using an iterative refinement algorithm in MUSCLE with default parameters [8] and manually edited in MEGA software version 11 [9]. The analysis of overall domain family of the BlaIMP was conducted in Pfam [10].

### 2.3. Phylogenetic Tree Estimation

Prior to the construction of the phylogenetic tree, the model test was conducted to estimate the most appropriate model using built-in functions in MEGA [9]. The maximum likelihood phylogenetic tree with 1000 bootstraps was constructed using General Time Reversible (GTR) model with gamma distribution for nucleotide sequences using FastTree [11]. The tree was visualised in FigTree (FigTree. Available online: http://tree.bio.ed.ac.uk/software/figtree/, accessed on 30 November 2021) and annotated in the interactive Tree of Life (iTOL) [12].

## 3. Results

### 3.1. Distribution of IMP-Type MBLs

A search of the NCBI database and EMBASE using “IMP and β-Lactamases” for gene encoding *bla_IMP_* demonstrated a variety of variants of IMP-type MBL genes as well as species of IMP-carrying organisms. There were 88 variants of IMP-type MBL genes currently deposited on NCBI’s GenBank. These 88 variants were identified in 29 species across 32 countries (Table 1). Interestingly, most of the *bla_IMP_* genes identified were from hospital isolates (Table 1). According to the genes submitted to GenBank and the literature search, the detection of *bla_IMP_* was frequently reported from Japan (25%), followed by China (17%) and France (7%) (Figure 1A).

According to Figure 1A, Asia accounted for 69% of the reporting countries. The presence of the *bla_IMP_* gene was reported in 12 countries, namely, China (including Hong Kong), India, Iran, Japan, Korea, Malaysia, Nepal, Singapore, Thailand, Turkey and Vietnam. Focusing on Asia, Japan and China were the first (36%) and the second (25%) most frequently *bla_IMP_* identified countries. Thailand and Singapore were the third most frequently reported countries (7%) (Figure 1B). The most frequently reported *bla_IMP_* carriers were *Pseudomonas aeruginosa*, followed by *Acinetobacter baumannii*, *Klebsiella pneumoniae* and *Enterobacter cloacae*. By considering the variant of *bla_IMP_* in countries with high prevalence of *bla_IMP_* in Asia, *bla_IMP−1_* was the most frequently reported in Japan (23%) and Singapore (50%). *bla_IMP−4_* and *bla_IMP−14_* were the most frequently reported in China (27%) and Thailand (27%), respectively (Figure 2A–D).

### 3.2. In Silico Analysis of IMP-Type MBLs

In silico analysis of IMP-type MBL genes was conducted to investigate the diversity of the genes. Using multiple sequence alignment of 88 variants of IMP-type MBLs, the conserved sequences of active sites were identified as follows: His95, Phe96, His97, Asp99, Ser100, His157, Cys176 and His215 (numbered according to IMP−1; Appendix A) [17]. These sequences were residues of a lactam ring-catalytic site. The overall analysis showed 79.3%–96.7% amino acid sequence similarity. To investigate other functions of the protein, we performed protein domain prediction in Pfam. The result showed that this protein contained only one domain, namely ‘Metallo-hydrolase-like-MBL-fold superfamily’, covering from amino acid position 23 to position 234 (result not shown).

A phylogenetic tree was constructed to visualise the relationship of the genes. *bla_IMP_* genes were separated into three main clusters (Figure 3). Group I contains 38 variants. Noticeably, *bla_IMP−12_*, *bla_IMP−63_* and *bla_IMP−90_*, previously identified as group II [18], were currently in a subgroup of group I, called group Ia, with 95.1% bootstrap support. These three variants were isolated from strains with European origin. Group II contains 41 variants. Lastly, group III contains nine variants (Figure 3).

### 3.3. Resistance of IMP MBL Variants-Carrying Strains

The pattern of antibiotic susceptibility of the isolate carrying each *bla_IMP_* variant was obtained from the articles to investigate whether the variation in each variant was associated with susceptibility. The susceptibility profiles were taken from bacterial isolates carrying those *bla_IMP_* genes. By reviewing the literature, most of the antibiotic agents tested were in the group of cephalosporin and carbapenem (Figure 3), especially anti-pseudomonal antibiotics, since *P*. *aeruginosa* was the most abundant species identified to possess the *bla_IMP_* gene. Out of 88 available variants, susceptibility profile was reported only in 32 variants (Figure 3, right panel). Overall, strains with *bla_IMP_* were resistant to several β-lactam antibiotics.

For carbapenem, almost all of the isolates with *bla_IMP_* variants were resistant to both meropenem and imipenem. IMP−19, −28, and −34 enzymes were unable to inactivate the carbapenems. Similarly, Cephalosporin was shown to be less active against *bla_IMP_* -carrying species. Likewise, isolates with *bla_IMP_* were resistant to cephalosporins. Aztreonam, a monobactam, was also shown to have an anti-bacterial effect on most of *bla_IMP_* carriers.

By combining the antibiotic susceptibility profile with the phylogenetic tree to investigate the relationship between clustering and susceptibility, it was found that susceptibility pattern was not associated with the phylogenetic tree (Figure 3).

## 4. Discussion

The attention to clinically important bacteria has been rising due to the multidrug resistance caused by the production of drug-inactivating enzymes, especially β-lactamases [19]. More critically, the carbapenemase enzyme has been increasingly detected in pathogens that are associated with nosocomial infections [20,21]. This study is the first to comprehensively investigate the epidemiology and the diversity of IMP-type MBLs, class B β-lactamase with carbapenemase ability.

An IMP-type MBL is encoded by the *bla_IMP_**_-_**_N_* gene (N = an order of variant discovered) which can be located on the chromosome or the plasmid, which facilitates the transfer of the *bla_IMP_* gene via horizontal gene transfer [22,23]. Our study showed that the *bla_IMP_* gene was detected in clinically relevant species, including *P*. *aeruginosa* and *A*. *baumannii*, which are associated with nosocomial infection and listed in “Priority 1: CRITICAL” list of antibiotic resistant pathogens by WHO (WHO publishes list of bacteria for which new antibiotics are urgently needed. Available online: https://www.who.int/news/item/27-02-2017-who-publishes-list-of-bacteria-for-which-new-antibiotics-are-urgently-needed, accessed on 30 December 2021). Interestingly, our analysis revealed that the top two countries where *bla_IMP_* genes were detected were both Asian countries: Japan and China. Japan is the first place where IMP-type MBLs (IMP−1) were reported [6]. In Asia, there were 28, 15, 7 and 5 variants of the *bla_IMP_* gene identified in Japan, China, Thailand and Singapore, respectively. A recent study revealed that carbapenemases (derived from *P*. *aeruginosa*) are distributed throughout Thailand [18]. However, the epidemiological study of IMP variants in Japan and China has not yet been conducted. It is, therefore, important to note that the *bla_IMP_* gene is one of the causes of antibiotic resistance in Asia.

The phylogenetic tree is commonly used to investigate the evolutionary relationship of genes or organisms. Our findings revealed that a reconstructed phylogenetic tree using 88 *bla_IMP_* variants clustered the genes into three main groups (Figure 2). In a broad picture, this tree was similar to the previous version [18]. Nevertheless, group Ia, which was previously clustered in group II, was currently identified in group I with high bootstraps. It is important to note that the structure of phylogeny of *bla_IMP_* is nearly well-defined although some branches remain dynamic depending on the number of genes added to the tree. The change of position in the phylogenetic tree could be caused by the increased number of tested genes in our study.

A search for antibiotic susceptibility profiles revealed that strains containing 32 variants (out of 88) were tested for their susceptibility. The profile showed that the 3rd-generation cephalosporins and carbapenem were less effective against most strains with the *bla_IMP_* gene. Interestingly, Aztreonam is the only agent that is active to the strains with most types of *bla_IMP_* (Figure 3). However, the association between susceptibility and the phylogenetic tree was absent. This is supported by the findings showing that the sequence of the active site (catalytic site) was highly conserved within the members of MBLs [17]. It is of note that nucleotide or amino acid substitutions outside the active site might not affect the β-lactam-hydrolysing activity of the enzyme. In addition, the susceptibility profile of the strains containing each *bla_IMP_* variant must be performed to ensure the association between the substitution/phylogenetic tree and the antibiotic resistance pattern. It is important to note that the susceptibility profile was taken from bacterial isolates, so the susceptibility can be affected by another mechanism, such as other β-lactamases or efflux pumps [24]. All in all, the findings of this work demonstrated that antibiotic resistance-associated genes were distributed to several regions around the world. This emphasised that the need of discovering or inventing novel antibiotic agents and enforcing antibiotic stewardship is urgent.

## 5. Conclusions

Carbapenemase, especially IMP-type MBLs, causes public health problems worldwide. This study is the first to comprehensively analyse all currently available variants of IMP-type MBLs and their associated susceptibility. Asian countries, especially Japan and China, are presently under a wide spread of *bla_IMP_* -carrying bacteria which are antibiotic-resistant organisms listed by WHO. An unrooted phylogenetic backbone of *bla_IMP_* gene variants illustrated two separate groups without susceptibility or geographical association. This strengthens antibiotic stewardship policy on a global level to control antibiotic resistance problems.

## Figures and Tables

**Figure 1 antibiotics-11-00236-f001:**
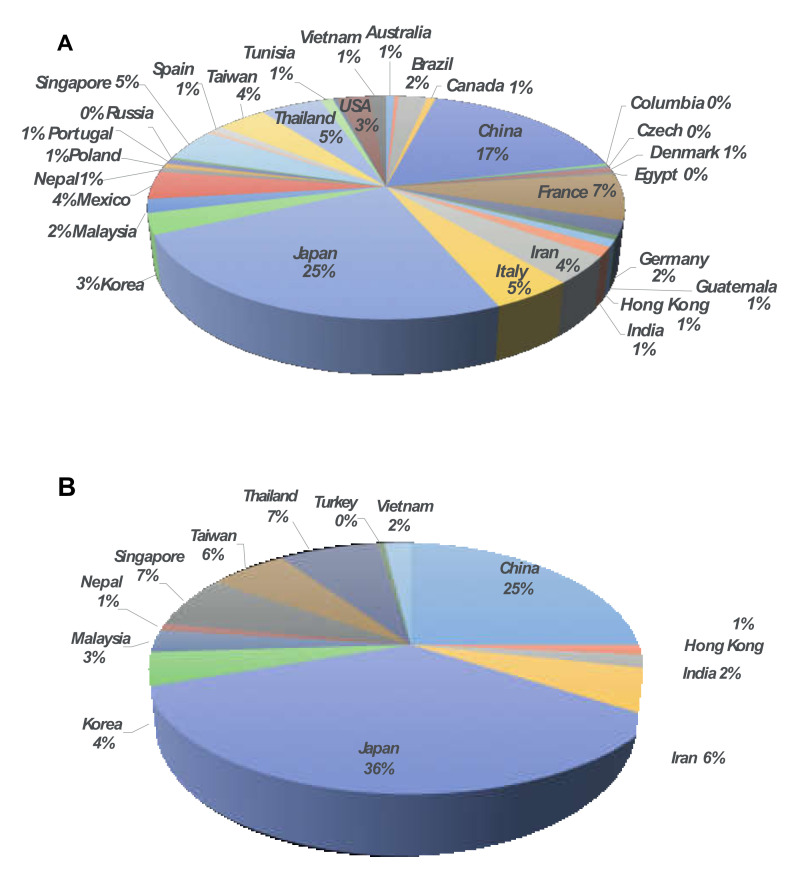
Distribution of IMP-type metallo-β-lactamase annotated genes (**A**) worldwide (**B**) in Asia.

**Figure 2 antibiotics-11-00236-f002:**
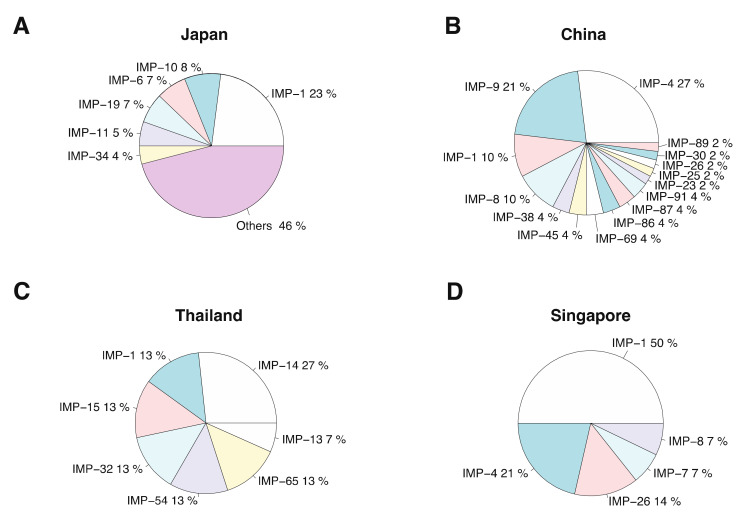
Distribution of *bla_IMP_* annotated genes in four countries in Asia: (**A**) Japan, (**B**) China, (**C**) Thailand and (**D**) Singapore. IMP-N is used to represent blaIMP-N.

**Figure 3 antibiotics-11-00236-f003:**
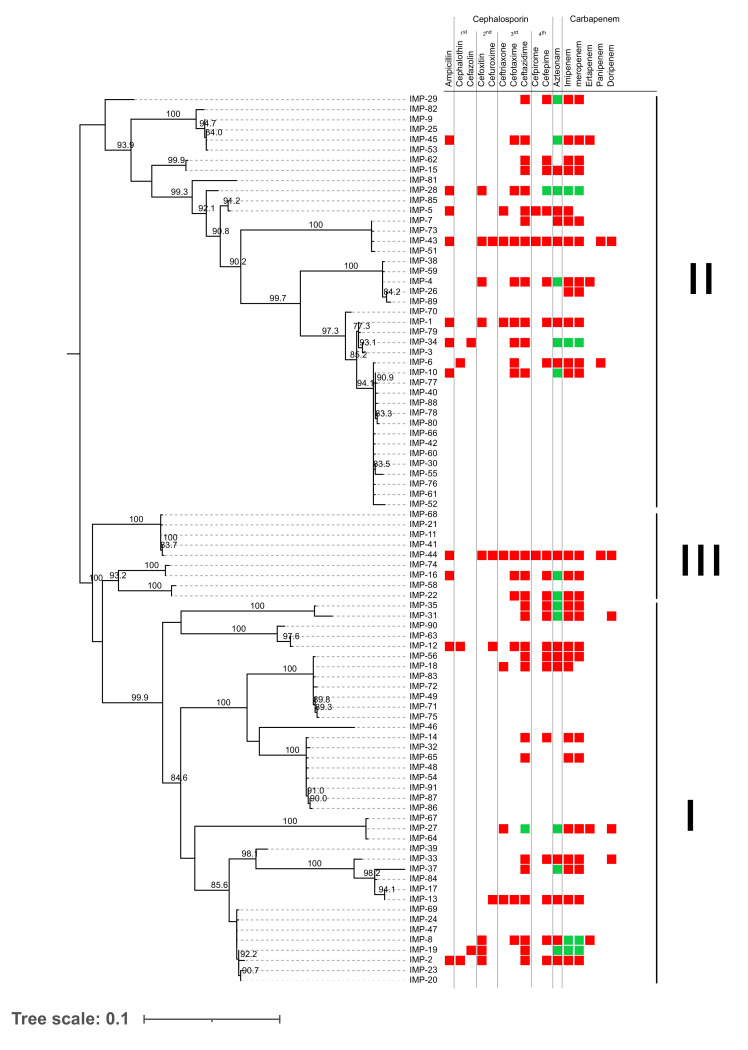
Phylogenetic relationship of *bla_IMP_* genes. An unrooted maximum likelihood phylogenetic tree constructed using nucleotide sequences of 88 *bla_IMP_* genes with 1000 bootstrap supports was visualised together with antibiotic susceptibility profile of 32 variants of the *bla_IMP_* gene. Red squares indicated “resistant” while green squares indicated “susceptible”. IMP-N is used to represent blaIMP-N.

**Table 1 antibiotics-11-00236-t001:** List of currently available IMP-type metallo-β-lactamase genes.

IMP Type	Host	Country of Isolation	Reference or Accession	Source of Isolates
**IMP−1**	*Achromobacter xylosoxidans*	Japan	EF027105.1, KF032823.1, KF032821.1, KF032820.1	Hospital
	*Comamonas thiooxydans*	Japan	AP025194.1	Hospital
	*Pseudomonas aeruginosa*	Japan	AB983593.1	Hospital
		Thailand	[13]	Hospital
		Malaysia	KX987869.1	Hospital
		China	AY386702.1, AY912485.1	Hospital
		Iran	KR703251.1, JX648311.1, JX644173.1, JQ766530.1	Hospital
		Nepal	LC636409.1	Hospital
		Singapore	AY168635.1, AY625689.1, AY625688.1, AY625687.1, AY625686.1	Hospital
		Egypt	KX452681.1	Hospital
		(Direct submission from Brazil)	GU831553.1, GU831552.1, GU831551.1, GU831550.1, GU831549.1, GU831548.1, GU831547.1, GU831546.1	N/A
		(Submitted from the UK, unpublished)	MH594579.1	Hospital
		Turkey	DQ842025.1	Hospital
		India	KF570107.1	Hospital
		USA	MK388919.1, MF479262.1	N/A
	*Pseudomonas putida*	Singapore	AY251052.1	Hospital
	*Pseudomonas fluorescens*	Singapore	AY250709.1	Hospital
	*Serratia marcescens*	Japan	AB162950.1, AB162949.1, AB162948.1, AB162947.1, NG_049172.1	Hospital
	*Klebsiella pneumoniae*	Iran	LC512050.1, LC512051.1	Hospital
	*Klebsiella pneumoniae*	Japan	[14]	Hospital
	*Acinetobacter* spp.	Korea	[15]	Hospital
	*Acinetobacter bereziniae*	Korea	EU014166.1, EU686386.1	Hospital
	*Acinetobacter calcoaceticus*	Thailand	HM185482.1	Hospital
	*Acinetobacter baumannii*	Japan	[15]	Hospital
		(Submitted from Korea, unpublished)	EF375699.1	Hospital
		Iran	KR080548.1, KF723585.1	Hospital
		(Submitted from Brazil, unpublished)	KF381490.1, KF381489.1, KF381488.1, KF381487.1	Hospital
		Thailand	HM036079.1	Hospital
	*Acinetobacter pittii*	Korea	GQ288398.1, GQ288393.1	Hospital
		Taiwan	GU064942.1, GU064941.1	N/A
			GQ864268.1	Hospital
		Japan	AB753459.1	N/A
	*Acinetobacter nosocomialis*	Korea	GQ288394.1	Hospital
		Taiwan	GU064940.1, GU064939.1, GU064938.1	N/A
	*Citrobacter freundii*	Japan	AB754498.1	N/A
	*Citrobacter youngae*	(Direct submission from Ireland)	MW847603.1	Hospital
	*Enterobacter aerogenes*	Japan	[15]	Hospital
	*Enterobacter cloacae*	(Direct submission from Japan)	LC508022.1	Hospital
		Japan	[15]	Hospital
		China	MK088089.1	Hospital
	*Enterobacter hormaechei*	China	MG287118.1	N/A
	*Escherichia coli*	Japan	[16]	Hospital
		Iran	LC512049.1	Hospital
	*Proteus mirabilis*	Brazil	KY057362.1	Hospital
	*Proteus vulgaris*	Japan	[16]	Hospital
	*Providencia rettgeri*	Japan	AB754496.1	N/A
	*Leclercia adecarboxylata*	China	KJ531212.1	Hospital
IMP−2	*Acinetobacter baumannii*	Italy	AJ243491.1, NG_049183.1	Hospital
		India	KC588963.1	Hospital
	*Serratia marcescens*	Japan	AB182996.1	N/A
	*Pseudomonas aeruginosa*	India	KC588963.1	Hospital
IMP−3	*Shigella flexneri*	(Published in USA)	NG_049194.1	N/A
IMP−4	*Acinetobacter baumannii*	Hong Kong	NG_049203.1, AF445082.1, AF244145.1	Hospital
		Singapore	DQ532122.1, AY795963.1, AY590475.1	Hospital
	*Acinetobacter calcoaceticus*	(Direct submission from Malaysia, unpublished)	DQ307573.1	N/A
	*Citrobacter freundii*	China	EU368857.1	Hospital
			JQ818252.1	N/A
	*Escherichia coli*	China	AB636651.1	N/A
		(Direct submission from India)	MF169878.1	N/A
	*Enterobacter cloacae*	China	KF699334.1	Hospital
		Korea	KY884003.1	N/A
		Japan	LC198842.1	Hospital
	*Enterobacter aerogenes*	China	KF184385.1	Hospital
	*Klebsiella pneumoniae*	China	EU368858.1, KF184388.1, FJ384365.1	Hospital
			JQ808503.1, JN106667.1, KF680003.1	N/A
	*Klebsiella oxytoca*	China	JQ820404.1	N/A
			KY913900.1	Animal
	*Pseudomonas aeruginosa*	China	DQ297664.1	N/A
		Malaysia	GQ221782.1	Hospital
IMP−5	*Acinetobacter baumannii*	Portugal	NG_049212.1, JF810083.1	Hospital
IMP−6	*Escherichia coli*	Japan	AB753460.1	N/A
	*Serratia marcescens*	Japan	NG_049220.1, AB040994.1	Hospital
	*Providencia rettgeri*	Japan	AB754497.1	N/A
	*Pseudomonas aeruginosa*	Japan	AB188812.1	Hospital
		Korea	EU117233.1	Hospital
IMP−7	*Pseudomonas aeruginosa*	Canada	NG_049221.1, AF318077.1	Hospital
		Czech	JX982232.1	Hospital
		Japan	LC091209.2, LC091210.2	Hospital
		Malaysia	GQ221781.1, AF416736.2, GU213192.1	Hospital
		India	HM641894.1	Hospital
		Singapore	AY625685.1	Hospital
		Slovakia	EF601914.1	Hospital
IMP−8	*Acinetobacter baumannii*	Taiwan	EF127959.1	Hospital
		China	DQ845788.1	Hospital
	*Escherichia coli*	Singapore	KF534724.1	Hospital
	*Enterobacter cloacae*	Taiwan	[16]	Hospital
		China	JQ820405.1	N/A
	*Klebsiella pneumoniae*	China	JQ820406.1, EU368856.1	Hospital
		Taiwan	NG_049222.1, AF322577.2	Hospital
		Tunisia	HE605039.1	Non-hospital
	*Klebsiella oxytoca*	China	HQ651093.1	Hospital
	*Serratia marcescens*	Taiwan	EU042136.1	N/A
IMP−9	*Pseudomonas aeruginosa*	China	AY033653, EU176818.1	Hospital
			KF184386.1, KF255597.1, KF255596.1, KF255595.1	N/A
		(Direct submission from China)	HM106459.1	N/A
IMP−10	*Achromobacter xylosoxidans*	Japan	AB074435.1, AB195638.1	Hospital
	*Pseudomonas aeruginosa*	Japan	AB074434.1, AB074433.1, NG_049173.1, AB195637.1	Hospital
		(Direct submission from Japan, Unpublished)	DQ288156.1	Hospital
	*Pseudomonas putida*	Italy	AJ420864.1	Hospital
	*Klebsiella pneumoniae*	Tunisia	HE605040.1	Non-hospital
IMP−11	*Pseudomonas aeruginosa*	Japan	AB074437.1	Hospital
	*Acinetobacter baumannii*	Japan	AB074436, NG_049174.1	Hospital
	*Enterobacter cloacae*	Japan	LC628821.1	N/A
IMP−12	*Pseudomonas putida*	Italy	NG_049175.1	Hospital
IMP−13	*Pseudomonas aeruginosa*	Italy	FJ172676.1, FJ172674.1, AJ512502.1, NG_049176.1	Hospital
		France	JX131371.1	Hospital
		Thailand	GU207399.1	Hospital
	*Pseudomonas monteilii*	Italy	JN091097.1	Hospital
	*Klebsiella pneumoniae*	Tunisia	HE605041.1	Non-hospital
IMP−14	*Achromobacter xylosoxidans*	Thailand	KJ406506.2, KJ406505.2	Hospital
	*Pseudomonas aeruginosa*	Thailand	AY553332.1, NG_049177.1	Hospital
IMP−15	*Pseudomonas aeruginosa*	Thailand	NG_049178.1, AY553333.1	Hospital
		Vietnam	LC075716.1	N/A
		Spain	KC310496.1	Hospital
IMP−16	*Pseudomonas aeruginosa*	Brazil	AJ584652.2, NG_049179.1	Hospital
IMP−17	*Pseudomonas aeruginosa*	Italy	NG_049180.1	Hospital
IMP−18	*Pseudomonas aeruginosa*	USA	AY780674.2, NG_049181.1	Hospital
		Mexico	HM138673.1	N/A
		(Direct submission from Costa Rica, unpublished)	KC907377.2	Hospital
		(Direct submission from Japan, unpublished)	AB587676.1	N/A
IMP−19	*Acinetobacter baumannii*	Iran	JQ766528.1	N/A
		Japan	AB184977.1	Hospital
	*Achromobacter xylosoxidans*	Japan	AB201263.1	N/A
	*Enterobacter cloacae*	Japan	AB201264.1	N/A
	*Aeromonas caviae*	France	NG_049182.1	Hospital
	*Klebsiella pneumoniae*	(Direct submission from Japan, unpublished)	LC062960.1	Hospital
	*Pseudomonas aeruginosa*	Japan	AB184976.1	Hospital
	*Pseudomonas putida*	Japan	AB201265.1	N/A
	*Serratia marcescens*	Poland	MH071810.1	N/A
			MF678587.1	Hospital
IMP−20	*Pseudomonas aeruginosa*	Japan	AB196988, NG_049184.1	N/A
IMP−21	*Pseudomonas aeruginosa*	Japan	AB204557, NG_049185.1	N/A
IMP−22	*Providencia rettgeri*	Japan	AB754495.1	N/A
	*Pseudomonas aeruginosa*	Austria	FM876313.1	Hospital
	*Pseudomonas fluorescens*	Italy	DQ361087.2, NG_049186.1	Non-hospital
IMP−23	*Citrobacter freundii*	China	NG_049187.1	N/A
IMP−24	*Serratia marcescens*	Taiwan	EF192154.1, NG_049188.1	Hospital
IMP−25	*Pseudomonas aeruginosa*	China	EU352796	Hospital
		Korea	EU541448.1, NG_049189.1	Hospital
		(Direct submission from China, unpublished)	KY081418.1, KY081417.1, HM175876.1	N/A
	*Stenotrophomonas maltophilia*	(Direct submission fom China)	GU944726.1	N/A
IMP−26	*Enterobacter cloacae*	China	HQ685900.1	Hospital
	*Pseudomonas aeruginosa*	Malaysia	JQ629930.1	Hospital
	*Pseudomonas aeruginosa*	Nepal	LC636067.1	Hospital
	*Pseudomonas aeruginosa*	Singapore	GU045307.1, NG_049190.1	Hospital
	*Pseudomonas aeruginosa*	Vietnam	LC075717.1	N/A
IMP−27	*Morganella morganii*	Mexico	KY847875.1, KY847873.1	N/A
	*Proteus mirabilis*	USA	JF894248.1	Hospital
		(Direct submission from USA)	NG_049191.1	N/A
	*Providencia rettgeri*	USA	KY847874.1	N/A
IMP−28	*Klebsiella oxytoca*	Spain	HQ263342.1, NG_049192.1	Hospital
IMP−29	*Pseudomonas aeruginosa*	France	HQ438058.1, JQ041634, NG_049193.1	Hospital
IMP−30	*Escherichia coli*	China	KM589497.1	Hospital
	*Pseudomonas aeruginosa*	Russia	NG_049195.1	N/A
IMP−31	*Pseudomonas aeruginosa*	Germany	KF148593.1, NG_049196.1	Hospital
IMP−32	*Klebsiella pneumoniae*	Thailand	NG_049197.1, JQ002629.1	Hospital
IMP−33	*Pseudomonas aeruginosa*	Italy	JN848782, NG_049198.1	Hospital
IMP−34	*Klebsiella oxytoca*	Japan	AB700341.1, NG_049199.1	Hospital
	*Acinetobacter colistiniresistens*	Japan	LC276939.1	Hospital
IMP−35	*Pseudomonas aeruginosa*	Germany	JF816544.1, NG_049200.1	Hospital
IMP−36	Not found in NCBI database and pubmed			
IMP−37	*Pseudomonas aeruginosa*	France	JX131372.1, NG_049201.1	Hospital
IMP−38	*Klebsiella pneumoniae*	China	HQ875573.1, NG_049202.1	N/A
IMP−39	*Pseudomonas aeruginosa*	France	MK507818.1, NG_064724.1	Hospital
IMP−40	*Pseudomonas aeruginosa*	Japan	AB753457, NG_049204.1	N/A
IMP−41	*Pseudomonas aeruginosa*	Japan	AB753458, NG_049205.1	N/A
IMP−42	*Acinetobacter soli*	Japan	AB753456.1, NG_049206.1	N/A
IMP−43	*Pseudomonas aeruginosa*	Japan	NG_049207.1	Hospital
IMP−44	*Pseudomonas aeruginosa*	Japan	NG_049208.1	Hospital
IMP−45	*Pseudomonas aeruginosa*	China	KJ510410.1, NG_049209.1	Animal
		France	KU984333.1	Hospital
IMP−46	*Pseudomonas putida*	France	MK543944.1, MK507819.1, NG_064725.1	Hospital
IMP−47	*Serratia marcescens*	(Direct submit USA)	KP050486.1	N/A
IMP−48	*Pseudomonas aeruginosa*	(Direct submit USA, unpublished)	NG_049210.1, KM087857.1	N/A
IMP−49	*Pseudomonas aeruginosa*	Brazil	NG_049211, KP681694.1	N/A
IMP−50	Not found in NCBI database and pubmed			
IMP−51	*Pseudomonas aeruginosa*	Vietnam	NG_049213.1, LC031883.1	Hospital
IMP−52	*Escherichia coli*	Japan	NG_049214.1, LC055762.1	N/A
IMP−53	*Pseudomonas aeruginosa*	(Direct submit USA)	NG_049215.1	N/A
IMP−54	*Pseudomonas aeruginosa*	Thailand	KU052795.1, NG_049216.1	N/A
IMP−55	*Acinetobacter baumannii*	Iran	KU299753.1, NG_049217.1	Hospital
IMP−56	*Pseudomonas aeruginosa*	Mexico	KU351745.1	Hospital
		Guatemala	KU315553.1, NG_049218.1	N/A
IMP−57	Not found in NCBI database and pubmed			
IMP−58	*Pseudomonas putida*	Denmark	KU647281.1, NG_049219.1	N/A
IMP−59	*Escherichia coli*	Australia	KX196782.1, NG_055477.1	N/A
IMP−60	*Enterobacter cloacae*	Japan	LC159227.1, NG_050945.1	Hospital
IMP−61	*Acinetobacter baumannii*	(Direct submission from Germany, unpublished)	KX462700.1, NG_051166.1	Hospital
IMP−62	*Pseudomonas aeruginosa*	Mexico	KX753224.1, NG_051513.1	Hospital
IMP−63	*Pseudomonas aeruginosa*	France	KX821663.1, NG_052049.1	Hospital
IMP−64	*Proteus mirabilis*	USA	NG_054710.1, KX949735.2	N/A
IMP−65	*Pseudomonas aeruginosa*	Thailand	KY315991.1, NG_066508.1	Hospital
IMP−66	*Escherichia coli*	Japan	LC190726.1, NG_054676.1	N/A
IMP−67	*Providencia rettgeri*	(Direct submission from USA, unpublished)	MF281100.1, NG_055271.1	N/A
IMP−68	*Klebsiella pneumoniae*	Japan	MF669572.1, NG_055584.1	N/A
IMP−69	*Providencia* spp.	China	MF678349.1, NG_055665.1	N/A
IMP−70	*Pseudomonas aeruginosa*	Germany	MG748725.1, NG_056176.1	Hospital
	*Providencia rettgeri*	Japan	LC348383.1	N/A
IMP−71	*Pseudomonas aeruginosa*	France	MG818167.1	Hospital
IMP−72	*Pseudomonas aeruginosa*	Mexico	MH021847.1	N/A
IMP−73	*Pseudomonas aeruginosa*	Japan	MH021848.1, NG_057463.1	N/A
IMP−74	*Pseudomonas aeruginosa*	Brazil	MH243349.1, NG_057606.1	N/A
IMP−75	*Pseudomonas aeruginosa*	Mexico	MH243350.1, MW692112.1, NG_057607.1	N/A
IMP−76	*Pseudomonas aeruginosa*	Japan	NG_061409.1	Hospital
IMP−77	*Pseudomonas aeruginosa*	Japan	NG_061410.1	Hospital
IMP−78	*Pseudomonas aeruginosa*	Japan	NG_061411.1	Hospital
IMP−79	*Pseudomonas aeruginosa*	France	MG873561.1, NG_061626.1	Hospital
IMP−80	*Pseudomonas aeruginosa*	Japan	NG_062274.1	Hospital
IMP−81	*Pseudomonas aeruginosa*	Columbia	MN267699.1	N/A
IMP−82	*Pseudomonas aeruginosa*	(Direct submission from Germany, unpublished)	MN057782.1	Hospital
		(Direct submission from USA, unpublished)	NG_065873.1	Hospital
IMP−83	*Pseudomonas aeruginosa*	Mexico	MN104595.1, NG_065874.1	N/A
IMP−84	*Pseudomonas aeruginosa*	(Direct submission from Switzerland, unpublished)	MN219692.1	N/A
	*Pseudomonas aeruginosa*	(Direct submission from USA, unpublished)	NG_065875.1	N/A
IMP−85	*Pseudomonas aeruginosa*	France	MN510335.1, NG_066696.1	Hospital
IMP−86	*Pseudomonas aeruginosa*	China	MT241520.1, NG_076650.1	N/A
IMP−87	*Pseudomonas aeruginosa*	China	MT241521.1, NG_076651.1	N/A
IMP−88	*Pseudomonas aeruginosa*	Japan	LC558310.1, NG_070737.1	Hospital
IMP−89	*Pseudomonas putida*	China	NG_070738.1	N/A
IMP−90	*Pseudomonas aeruginosa*	(Direct submission from Germany, unpublished)	MW811441.1	Hospital
		(Direct submission from USA, unpublished)	NG_074713.1	Hospital
IMP−91	*Pseudomonas aeruginosa*	China	MZ702721.1, NG_076634.1	N/A

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
