# Peer review of "Comprehensive Analysis of Imipenemase (IMP)-Type Metallo-β-Lactamase: A Global Distribution Threatening Asia"

_antibiotics, 2022, doi:10.3390/antibiotics11020236_

Round 1
Reviewer 1 Report
Dear Autors,
The manuscript submitted for review raises a topic that worries people around the world, namely bacterial resistance to antibiotics (beta-lactams) used in therapy.
However, there are glaring errors in the manuscript: - Table 1 - why the names of bacteria are not standardized? Once the name isfull, other times it is abbreviated. Please standardize - incorrectly used name. We write Acinetobacter baumannii using two "i" at
the end. - "spp" is not written in italics The icing on the cake would be to show the strains analyzed by the authors
in terms of their presence: hospital / non-hospital. This would give us an
overall picture of where these resistant strains are more common.
Best regards.
Author Response
The manuscript submitted for review raises a topic that worries people around the world, namely bacterial resistance to antibiotics (beta-lactams) used in therapy.
Response: We thank the Reviewer for the constructive comments and suggestions which we feel help improving the manuscript significantly. Please find below the point-by-point responses to the Reviewer’s comments. All changes have been labeled as the red colour text and the track change provided in the revised manuscript already.
We have edited the title of the manuscript, changed from “Comprehensive analysis of imipenemase (IMP)-type metallo-Beta-lactamase showing global distribution threating Asia” into “Comprehensive analysis of imipenemase (IMP)-type metallo-β-lactamase: a global distribution threatening Asia”.
Point 1: However, there are glaring errors in the manuscript: - Table 1 - why the names of bacteria are not standardized? Once the name is
full, other times it is abbreviated. Please standardize - incorrectly used name. We write Acinetobacter baumannii using two "i" at
the end. - "spp" is not written in italics The icing on the cake would be to show the strains analyzed by the authors
Response: Thank you for pointing out this important issue. We checked and corrected typographical errors throughout the manuscript.
Point 2: in terms of their presence: hospital / non-hospital. This would give us an
overall picture of where these resistant strains are more common.
Best regards.
Response: This is a very constructive comment. We have searched for more information about the origin of the blaIMP carrying isolates and appended the findings in the last column of table 1.
Reviewer 2 Report
The manuscript represents the analysis of open access scientific data about the distribution of bacterial genes encoding impemenase-type metallo-β-lactamases. Such a comprehensive review of these enzymes has not previously been performed.
1) Suggestions for the substance of the work
The only question concerns the conclusion that aztreonam “was also shown to have less effect on bla IMP carriers” (lines 138-139). First of all, it is not very clear what the antibiotic was less effective compared to. Other antibiotics or other bacterial strains (missing bla IMP genes)? There is no confirmation of any possibility in the text or the Figure. Vice versa, among all antibiotics analyzed in the manuscript, aztreonam proved to be the most effective against strains carrying IMP. I think attention should be paid to this finding in the text.
The authors predictably found marked phylogenetic conservation in amino acid residues constituting the catalytic site of IMP-type β-lactamases. In my opinion, it would be more interesting to find some conserved motifs outside the catalytic site and discuss their possible functional significance. Therefore, I suggest that the text of the manuscript be supplemented with such an analysis.
2) Comments on the quality of the text and data presentation
First of all, extensive editing of English language is required. Although I don't feel qualified in English and could be wrong myself in some instances listed below, I can see serious problems here.
Lines 34-35: class A, C, D are serine protease-derived β-lactamases while class B is the metallo-or zinc dependent β-lactamase (“include” or similar verb should be used here)
Line 52: Review of literatures (should be “literature”)
Line 90: Japan and China remained the first (36%) and the second (25%) 90 most frequently bla IMP identified countries (wrong verb)
Line 103: By using multiple sequence alignment… (Just “Using…”)
Even the country names in Table 1 are sometimes misspelt (“Franch” and “German” instead of “France” and “Germany”).
Throughout the text, authors write about antibiotic susceptibility or resistance of genes and enzymes, whereas there should be bacterial strains carrying these genes/enzymes.
In Fig. 3 it should specify the meaning of the colour indicators (red and green) for antibiotic sensitivity, and title the antibiotic groups.
3) Minor issues
Line 105: These sequences were residues of a lactam ring-catalytic site
Line 114: IMP-12, IMP-63 and IMP-90, previously identified as group II
References are needed to confirm these claims.
Line 87 – something should be inserted in the place of the ellipsis.
Line 46: previous phylogenetic construction was restricted due to the number of available sequences
Perhaps, “due to the limited number” is better.
Captions to Fig. 1 and 2 – perhaps, “annotated genes” is better than “genes”.
Unfortunate phrase:
Line 144: The importance of clinically important bacteria
Author Response
The manuscript represents the analysis of open access scientific data about the distribution of bacterial genes encoding impemenase-type metallo-β-lactamases. Such a comprehensive review of these enzymes has not previously been performed.
Response: We thank the Reviewer for the constructive comments and suggestions which we feel help improving the manuscript significantly. Please find below the point-by-point responses to the Reviewer’s comments. All changes have been labeled as the red colour text and the track change provided in the revised manuscript already.
We have edited the title of the manuscript, changed from “Comprehensive analysis of imipenemase (IMP)-type metallo-Beta-lactamase showing global distribution threating Asia” into “Comprehensive analysis of imipenemase (IMP)-type metallo-β-lactamase: a global distribution threatening Asia”.
1) Suggestions for the substance of the work
Point1: The only question concerns the conclusion that aztreonam “was also shown to have less effect on bla IMP carriers” (lines 138-139). First of all, it is not very clear what the antibiotic was less effective compared to. Other antibiotics or other bacterial strains (missing bla IMP genes)? There is no confirmation of any possibility in the text or the Figure. Vice versa, among all antibiotics analyzed in the manuscript, aztreonam proved to be the most effective against strains carrying IMP. I think attention should be paid to this finding in the text.
Response 1: We would like to thank the Reviewer for mentioning this sentence. I have changed the phrase from “less effect” into “effect” since it was initially meant that Aztreonam was not active against all of blaIMP carriers.
Point 2: The authors predictably found marked phylogenetic conservation in amino acid residues constituting the catalytic site of IMP-type β-lactamases. In my opinion, it would be more interesting to find some conserved motifs outside the catalytic site and discuss their possible functional significance. Therefore, I suggest that the text of the manuscript be supplemented with such an analysis.
Response 2: We appreciate the Reviewer’s comment regarding this point. The search for a predicted domain outside the catalytic site was done using Pfam. It revealed that this protein contained only one domain, called Metallo-hydrolase-like-MBL-fold superfamily, covering from amino acid position 23 to position 234 (There are 246 amino acids in this protein). Accordingly, the result of the analysis has been included in the Results section as follow;
Results; page 15, line 184-187:
To investigate other functions of the protein, we performed protein domain prediction in Pfam. The result showed that this protein contained only one domain, namely ‘Metallo-hydrolase-like-MBL-fold superfamily’,covering from amino acid position 23 to position 234 (result not shown).
2) Comments on the quality of the text and data presentation
Point 3: First of all, extensive editing of English language is required. Although I don't feel qualified in English and could be wrong myself in some instances listed below, I can see serious problems here.
Response 3: We would like to thank the Reviewer for suggesting this. We asked professional English proofreader to edit the writing style and grammatical errors throughout this manuscript.
Point 4: Lines 34-35: class A, C, D are serine protease-derived β-lactamases while class B is the metallo-or zinc dependent β-lactamase (“include” or similar verb should be used here)
Response 4: We thank the reviewer for mentioning this point. We have re-written these sentences as follow:
Introduction; page 1, line 34-35:
Class A, C, D include serine protease-derived β-lactamases while class B includes the metallo-or zinc dependent β-lactamase (MBL)
Point 5: Line 52: Review of literatures (should be “literature”)
Response 5: We appreciate the Reviewer’s suggestion and have changed it accordingly.
Point 6: Line 90: Japan and China remained the first (36%) and the second (25%) 90 most frequently bla IMP identified countries (wrong verb)
Response 6: We appreciate the Reviewer’s suggestion.We have changed the word from “remained” into “were”.
Point 7: Line 103: By using multiple sequence alignment… (Just “Using…”)
Response 7: We appreciate the Reviewer’s suggestion and have changed it accordingly.
Point 8: Even the country names in Table 1 are sometimes misspelt (“Franch” and “German” instead of “France” and “Germany”).
Response 8: We appreciate the Reviewer’s suggestion. We have edited and corrected the misspelling of the names of the countries in Table 1.
Point 9: Throughout the text, authors write about antibiotic susceptibility or resistance of genes and enzymes, whereas there should be bacterial strains carrying these genes/enzymes.
Response 9: We thank the Reviewer for mentioning this issue. We have revised the susceptibility section and carefully amended accordingly.
Point 10; In Fig. 3 it should specify the meaning of the colour indicators (red and green) for antibiotic sensitivity, and title the antibiotic groups.
Response 10: We appreciate the Reviewer’s suggestion and have amended accordingly.
3) Minor issues
Point 11: Line 105: These sequences were residues of a lactam ring-catalytic site
Response 11: We appreciate the Reviewer’s suggestion and have amended accordingly.
Point 12: Line 114: IMP-12, IMP-63 and IMP-90, previously identified as group II
References are needed to confirm these claims.
Response 12: We thank the Reviewer for suggesting this point. We have added the reference that we use to claim this point.
Point 13: Line 87 – something should be inserted in the place of the ellipsis.
Response 13: We are grateful for the Reviewer’s comment. We have removed the incompleteness.
Point 14: Line 46: previous phylogenetic construction was restricted due to the number of available sequences
Perhaps, “due to the limited number” is better.
Response 14: We appreciate the Reviewer’s suggestion and have changed “the number” into “the limited number”
Point 15: Captions to Fig. 1 and 2 – perhaps, “annotated genes” is better than “genes”.
Response 15: We appreciate the Reviewer’s comment regarding this usage. As suggested, we have amended accordingly.
Unfortunate phrase:
Point 16: Line 144: The importance of clinically important bacteria
Response 16: We thank the Reviewer for spotting this error. We have changed “The importance of clinically important bacteria …” into “The attention to clinically important bacteria …”
Reviewer 3 Report
In this manuscript, the authors analyzed and compare all of the IMP metallo-beta-lactamases described to date. They report the Genus and species that the IMPs are found in as well as global and regional distribution. The further compare the blaIMP gene sequences, which cluster into two groups and also report available susceptibility data. The report is a valuable update in the IMP metallo-beta-lactamases, is comprehensive in nature, and well written.
Minor comments for consideration:
- Introduction vs. abstract vs. text have different number of IMPs 88 vs 91. The analysis was conducted on the 88? But 91 have been described up to the time of manuscript submission. Please provide some clarity to the average reader.
- Please check italicization of Pseudomonas aeruginosa throughout.
- Please use caution when using IMP-1 vs. blaIMP-1 as one is the protein and other is gene, respectively. In Figure 3, the gene sequences were compared correct; protein names are used in the Figure? The groupings may be different if protein sequences were used as silent mutations in genes would lead to same amino acid sequence. Please indicate in the legend what the green vs red squares represent.
- In terms of the susceptibility data available for the 32 variants, were these clinical isolates or were these data from the blaIMP cloned and expressed in a clean background or both. Please indicate in the text as again other resistance mechanisms in clinical isolates would impact the susceptibility data, thus making it more difficult to identify patterns based on phylogenetic groupings.
Author Response
In this manuscript, the authors analyzed and compare all of the IMP metallo-beta-lactamases described to date. They report the Genus and species that the IMPs are found in as well as global and regional distribution. The further compare the blaIMP gene sequences, which cluster into two groups and also report available susceptibility data. The report is a valuable update in the IMP metallo-beta-lactamases, is comprehensive in nature, and well written.
Response: We thank the Reviewer for the constructive comments and suggestions which we feel help improving the manuscript significantly. Please find below the point-by-point responses to the Reviewer’s comments. All changes have been labeled as the red colour text and the track change provided in the revised manuscript already.
We have edited the title of the manuscript, changed from “Comprehensive analysis of imipenemase (IMP)-type metallo-Beta-lactamase showing global distribution threating Asia” into “Comprehensive analysis of imipenemase (IMP)-type metallo-β-lactamase: a global distribution threatening Asia”.
Minor comments for consideration:
Point 1: Introduction vs. abstract vs. text have different number of IMPs 88 vs 91. The analysis was conducted on the 88? But 91 have been described up to the time of manuscript submission. Please provide some clarity to the average reader.
Response 1: We thank the Reviewer for mentioning this point. At the time of analysis, the nomenclature of the variant is up to blaIMP-91. When we analysed, the actual number of variants was 88. blaIMP-36, blaIMP-50 and blaIMP-57 could not be found in any database. Therefore, we have replaced “91” with “88”.
Point 2: Please check italicization of Pseudomonas aeruginosa throughout.
Response 2: We appreciate the Reviewer’s suggestion. We have checked the name of the species and make them into proper scientific nomenclature (italicised).
Point 3: Please use caution when using IMP-1 vs. blaIMP-1 as one is the protein and other is gene, respectively. In Figure 3, the gene sequences were compared correct; protein names are used in the Figure? The groupings may be different if protein sequences were used as silent mutations in genes would lead to same amino acid sequence. Please indicate in the legend what the green vs red squares represent.
Response 3: We appreciate the Reviewer’s suggestion concerning this point. For the phylogenetic tree, we compared nucleotide sequences of “blaIMP” gene and we used “IMP” to represent the variant of the gene (instead of writing the full name). We have added the sentence “IMP-N is used to represent blaIMP-N.” at the end of figure legend of Figure 3.
and have amended accordingly.
Point 4: In terms of the susceptibility data available for the 32 variants, were these clinical isolates or were these data from the blaIMP cloned and expressed in a clean background or both. Please indicate in the text as again other resistance mechanisms in clinical isolates would impact the susceptibility data, thus making it more difficult to identify patterns based on phylogenetic groupings.
Response 4: We thank the Reviewer for referring this point. The information was taken from blaIMP-carrying clinical isolates that were available in research articles. As suggested, we also mention that this information was taken from bacterial isolates. Other the susceptibility could be affected by other mechanisms.
Reviewer 4 Report
The authors described the distribution of IMP-type MBLs in Asia and raised the awareness of the situation of antimicrobial resistance in the region. I think that this work is good and suitable for publication after major revision, particularly the implementation of figures both for quality and type of representation.
English editing is needed, and some corrections througout all manuscript:
line 42: P. aeruginosa in italic
Beta...change in the symbol
line 87: ...????
Lines 109-122: do not report the names of IMP variants, there is the figure that describes
lane 177: some types...describe!!
lane 189: has caused...causes
Author Response
The authors described the distribution of IMP-type MBLs in Asia and raised the awareness of the situation of antimicrobial resistance in the region. I think that this work is good and suitable for publication after major revision, particularly the implementation of figures both for quality and type of representation.
English editing is needed, and some corrections througout all manuscript:
Response: We thank the Reviewer for the constructive comments and suggestions which we feel help improving the manuscript significantly. Please find below the point-by-point responses to the Reviewer’s comments. All changes have been labeled as the red colour text and the track change provided in the revised manuscript already.
We have edited the title of the manuscript, changed from “Comprehensive analysis of imipenemase (IMP)-type metallo-Beta-lactamase showing global distribution threating Asia” into “Comprehensive analysis of imipenemase (IMP)-type metallo-β-lactamase: a global distribution threatening Asia”.
Point 1: line 42: P. aeruginosa in italic
Response 1: We appreciate the Reviewer’s suggestion. We have checked the name of the species and make them into proper scientific nomenclature (italicised).
Point 2: Beta...change in the symbol
Response 2: We thank the Reviewer for mentioning this point. We have replaced “beta” with “β”.
Point 3: line 87: ...????
Response 3: We are grateful for the Reviewer’s comment. We have removed the incompleteness.
Point 4: Lines 109-122: do not report the names of IMP variants, there is the figure that describes
Response 4: We would like to thank the Reviewer for suggesting this point. We have removed the name of the variant and referred the Figure instead.
Point 5: lane 177: some types...describe!!
Response 5: We are grateful for the Reviewer’s comment. We have added in-text referral to the figure 3.
Point 6: lane 189: has caused...causes
Response 6: We are grateful for the Reviewer’s suggestion. We have changed from “has caused” into “causes”
Round 2
Reviewer 1 Report
Dear Authors,thank you very much for the corrections which, in my opinion, increased the
value of this manuscript. I have no more comments.
Good luck
Author Response
Point 1: thank you very much for the corrections which, in my opinion, increased the value of this manuscript. I have no more comments. Good luck
Response 1: We appreciate that the Reviewer finds this study interesting, novel and original. We are grateful for your positive comments and kind support.
Reviewer 2 Report
The manuscript has improved considerably by revision. It could be accepted after minor corrections.
Line 19-20: should be re-written so that antibiotic susceptibility/resistance relates to bacteria but not enzymes
Line 70: correct reference (Kumar, 2018) – should be [9]
Line 103: it seems that blaIMP-4 should be instead of IMP-4
Line 108: add the same note as in line 129 (“IMP-N is used to represent bla IMP-N”)
Line 113: add reference after “These sequences were residues of a lactam ring-catalytic site” (probably, [25] in current numbering)
Line 143: it seems there should be “have no effect”
Author Response
The manuscript has improved considerably by revision. It could be accepted after minor corrections.
Response: We thank the Reviewer for the constructive comments and helpful suggestions
which we feel help improving the quality of the manuscript significantly. Please find below
the point-by-point responses to the Reviewer’s comments. All changes have been labeled and tracked as
the red color text in the revised manuscript already.
Point 1: Line 19-20: should be re-written so that antibiotic susceptibility/resistance relates to bacteria but not enzymes
Response 1: We thank the Reviewer for concerning this point. We have amended the sentence by referring to bacterial strains containing the gene.
Point 2: Line 70: correct reference (Kumar, 2018) – should be [9]
Response 2: We thank the Reviewer for mentioning this typographical error. We have replaced “(Kumar, 2018)” with [9].
Point 3: Line 103: it seems that blaIMP-4 should be instead of IMP-4
Response 3: We appreciate the Reviewer’s comment concerning this point. We have already replaced “IMP-4” with “blaIMP-14”.
Point 4: Line 108: add the same note as in line 129 (“IMP-N is used to represent bla IMP-N”)
Response 4: We thank the Reviewer for mentioning this. We have amended the sentence “IMP-N is used to represent blaIMP-N.” in the figure legend of Figure 2 (as done in Figure 3).
Point 5: Line 113: add reference after “These sequences were residues of a lactam ring-catalytic site” (probably, [25] in current numbering)
Response 5: We thank the Reviewer comment on this point. We have added the reference after such a sentence.
Point 6: Line 143: it seems there should be “have no effect”
Response 6: We appreciate the Reviewer’s comment on this point. According to Figure 3, Aztreonam is the only antibiotic that can kill the strains with blaIMP gene (represented by “Green” colour in the Figure 3) (Red = resistant to the drug; Green = susceptible to the drug). We, therefore, replaced “have an effect” with “have an anti-bacterial effect”
Reviewer 4 Report
Thank you for your responses.
Author Response
We are glad that the Reviewer finds this manuscript is well-written and
provides useful information to support antibiotic smart use and stewardship in the future. We are grateful for your kind remarks and support.